# An anomaly detection scheme for data stream in cold chain logistics

Zhibo Xie[1]*, Heng Long[1], Chengyi Ling[1], Yingjun Zhou[2], Yan Luo[2]

1 School of information and intelligent engineering, Zhejiang wanli University, Ningbo, China, 2 Ningbo Municipal Bereau of Ecology and Enviroment, Ningbo, China

* xiezhibo@zwu.edu.cn

## Abstract

Anomaly detection is widely used in cold chain logistics (CCL). But, because of the high cost and technical problem, the anomaly detection performance is poor, and the anomaly can not be detected in time, which affects the quality of goods. To solve these problems, the paper presents a new anomaly detection scheme for CCL. At first, the characteristics of the collected data of CCL are analyzed, the mathematical model of data flow is established, and the sliding window and correlation coefficient are defined. Then the abnormal events in CCL are summarized, and three types of abnormal judgment conditions based on cor-relation coefficient $\rho_{jk}$ are deduced. A measurement anomaly detection algorithm based on the improved isolated forest algorithm is proposed. Subsampling and cross factor are designed and used to overcome the shortcomings of the isolated forest algorithm (iForest). Experiments have shown that as the dimensionality of the data increases, the performance indicators of the new scheme, such as $P$ (precision), $R$ (recall), $F1$ score, and AUC (area under the curve), become increasingly superior to commonly used support vector machines (SVM), local outlier factors (LOF), and iForests. Its average $P$ is 0.8784, average $R$ is 0.8731, average F1 score is 0.8639, and average AUC is 0.9064. However, the execution time of the improved algorithm is slightly longer than that of the iForest.

## Introduction

CCL is a supply logistics chain which uses refrigeration technology to maintain a suitable environment for those perishable products such as fruits,vegetables, dairy, meats, fish and medicine and so on [1]. With the improvement of living standards, people have higher requirements for food quality, CCL is more widely in food transportation. As the main way to ensure food quality, CCL anomaly detection has become more important. Unlike other anomaly detection techniques, CCL anomaly detection technology faces many challenges, such as data stream being in strong noise and interference environments; The measurement error will be large due to the sensor being in a low-temperature environment; The accuracy will decrease due to long-term use of the sensor; Real time requirements are high, but the amount of data generated during the monitoring process is huge and the data dimension is high, and so on. Therefore, although there are so many anomaly detection algorithms, there are few effective algorithms for CCL anomaly detection, and the performance of the currently used algorithms is poor. So it is necessary to develop efficient algorithms for CCL.

**Data availability statement:** All relevant data are within the manuscript and its Supporting

Information files. I have uploaded three datasets in supporting information.

**Funding:** This research was funded by Commonweal Projects of Zhejiang Province (Grant No. LGN20F010001) and Innovation and Entrepreneurship Project for College Students (Grant No. 202310876023 and s202310876033) awarded to ZX, HL and CL. The funders had no role in study design, data collection and analysis, decision to publish, or preparation of the manuscript.

**Competing interests:** The authors have declared that no competing interests exist.

A safety warning system is designed for a warehouse with a wireless communication network and multiple sensors to monitor the surrounding conditions such as fire and burglary [2]. The paper considers the instantaneous value of the sensor. As soon as the instantaneous value exceeds the threshold value, an alarm is given. Mariusz developed the theoretical basis for a rack technical parameter monitoring program to ensure structural reliability and avoid potential collisions in high compartments or high storage warehouses [3]. The paper mainly focuses on pressure and collision, and does not study other CCL parameters. A monitoring system based on multi-dimensional sensing is proposed in the paper [4]. But the sampling data must be read through a dedicated reader, so the hardware cost of the system is too high. Aiming at the quality problem in the CCL of dairy products, a new method of CCL pre-warning for dairy products based on support vector machine was proposed [5]. But the paper does not consider that the sampled data has the characteristics of data flow. Wang mentioned that smart tags can be used to detect food quality changes in CCL detection, but did not give a detailed hardware and software implementation [6].

In [7], a fresh food sensory perceptual system is designed for CCL, the cost is too high because of the specialized website and remote server. Wang S.X. researches the Tilapia Cold-chain Logistics abnormal temperature detection method, the structure of fuzzy ARMM was introduced in abnormal detection. The experimental results show that the proposed method to test the tilapia abnormal temperature of CCL, and overall performance are higher accuracy [8]. Feng researched a real-time monitoring system for fruit and vegetable CCL with ZigBee technology [9]. Liu studied the phase change cold storage materials in the CCL [10]. Witjaksono designed a temperature warning system, which can be triggered and send signals to the refrigeration system to adjust the refrigeration ventilation and keep the temperature in the suitable range [11]. A cold chain database platform based on network is developed for collecting and managing real-time temperature data to optimize and improve weak links in the supply chain [12]. Azzi et al. utilized blockchain technology to achieve distributed storage and management of supply chain data, to ensure data integrity, accuracy, and security [13]. An intelligent route-planning system based on IoT technology is designed by Tsang et al. The system is configured a WSN to monitor the entire chain in real-time to ensure the stable temperature during transportation [14]. Han predicted the forced air cooling efficiency of fresh apples by combining the optimal differential evolution algorithm with backpropagation neural network [15]. Neural network models require a large amount of data for pre training, and the amount of data will affect the efficiency of monitoring and analysis. The method of using thermal imaging to achieve two-dimensional images with multi-point temperature sensing has been proposed, while its accuracy is affected by the reflectivity of the monitored object [16–18]. The statistical process control and sensor networks is combined to used to detect and control cold chain temperature [19].

Hoang suggests that sustainable environmental control of refrigeration can be achieved by predicting future temperature changes [20]. The method of combining real-time environmental monitoring of the Internet of Things with early risk warning decision support systems to reduce food losses [14,21]. R. Jedermann found that local offset from the average value are much larger than expected when sensor networks are applied in containers [22].

Anomaly detection algorithms are generally divided into three kinds: unsupervised detection, semi-supervised detection and supervised detection [23]. If data labels can be obtained, supervised anomaly detection is preferred. KNN (K-Nearest-Neighbors), SVM (Support Vector Machine) are typical supervised detection algorithms. When there are only a few data labels, the semi- supervised anomaly detection model can also be used. But in fact, anomaly detection is often unlabeled, and the training data does not indicate which are

abnormal points, so unsupervised detection should be used. Principal Component Analysis (PCA), one class SVM, Angle-Based Outlier Detection (ABOD), LOF (Local Outlier Factor), isolated Forest are the main models of unsupervised detection [24–26]. Jie Tang proposed a supervies mechine model with SVM to control the CCL [27]. Wei Wu researched a platform with generative adversarial networks (GAN) and digital twin for CCL [28], which can be used for accident identification and indoor localization based on Bluetooch Low Energy to actualize real-time staff safety supervision in the cold warehouse. X. M. designed a cost-effective over-temperature alarm system using an artificial neural network model [29]. A unsupervised deep neural structure of stacked auto-encoder (SAE) was designed to identify abnormal stationary from human motion status [30]. One-dimensional point monitoring is currently the main way to measure the temperature and humidity of the cold chain envi ronment, and the results provide important parameters for evaluating whether the temperature and humidity meet the requirements of fresh produce and whether the food quality and safety are maintained [31]. This method of judging food-safety is one-sided, subjective, and unscientific due to problems such as a limited number of sensors and their impre cision, the uneven distribution and/or fluctuation of temperature and humidity in all stages of the cold chain, and the temperature and hu midity gradient that exists between the food and the environment, especially for packed fruits. Abdella et al. [32] and Badia-Melis et al. [33] used ANNs for the data correction and time series predic tion of single-point sensors for food CCL. The future temperature trends and demand disturbances of the cold chain have been accurately determined by back propa gation and deep learning neural networks (long short term memory [LSTM], stacked LSTM, bidirectional LSTM, convolutional LSTM), which have even replaced active RFID tags [20]. The forced air-cooling abnomal for fresh apples has been predicted by combining the optimal differential evolution algorithm and the back propagation neural network [15]. The main shortages of anomaly detection algorithms for CCL in the above documents are: (i) The particularity of CCL data acquisition is not considered, that is, it is usually necessary to arrange multiple sensors at different positions of a vehicle to make the measurement results more accurate. The data collected by each sensor, to be exact, should be data streams. Moreover, according to the actual situation, multiple sensors are often arranged in a CCL carriage. Taking temperature sensors as an example, generally at least 5 or more temperature sensors are arranged in different positions to obtain more accurate temperature. Therefore, CCL's data stream is a high-dimensional real-time data stream. (ii) The anomaly detection algorithms mentioned above are all focused on the same type of sensor, such as temperature, and do not include all commonly used sensors in CCL vehicles, such as temperature, humidity, oxygen concentration, carbon dioxide concentration, and pressure values. (iii) The multidimensional sensor data streams have strong temporal and spatial correlations, as well as strong noise.

In this paper, a novel CCL anomaly detection scheme is proposed, which not only considers the characteristics of data flow of the collected data of CCL, but also comprehensively considers the anomaly detection of multiple types of data. Firstly, the characteristics of the collected data of CCL are analyzed, the mathematical model of data flow is established, and the sliding window $|W|$ and correlation coefficient $\rho_{jk}$ are defined. Then the abnormal events in CCL are summarized, and three types of abnormal judgment conditions based on correlation coefficient $\rho_{jk}$ are deduced. Subsampling and cross factor are designed and used to overcome the shortcomings of the isolated forest (iForest) algorithm in detecting outliers with too many samples and too high dimensions. Experimental analysis shows positive results and demonstrates the effectiveness of designed CCL abnormal detection algorithm, compared with the state-of-art algorithm. Three main contributions of this paper can be summarized as follows:

(i) The characteristics of CCL multi-dimensional sensor data streams are analyzed, and mathematical analysis models for CCL data flow are deduced and established.

(ii) Three kinds of abnormal conditions in CCL are analyzed in detail. Based on the correlation characteristics of multi-dimensional data flow, the judgment scheme of each anomaly is given.

(iii) An improved isolated forest algorithm is proposed for the most common abnormal situation. The crossover factor is used to build new iTree and new forest. Compared with the common algorithms such as SVM, LOF and iForest, experiments show that the improved algorithm has better *P*, *R*, *F1* score and AUC without increasing the computational complexity, and the executing time is shorter.

The remaining paper is organized as follows. Section 2 details the proposed algorithm. Section 3 shows experimental results. Section 4 gives discussion. Finally, summary is given in Section 5.

## Materials and methods

The flowchart of the proposed algorithm, including training and testing, is shown in **Fig 1**. Flowchart of abnormal detection for CCL.This section introduces the proposed scheme in detail.

### Analysis and modeling of multi-sensor data stream

In the CCL environment, sensor data is a typical data flow. The so-called data stream, also known as stream data, refers to a data sequence that can only be read once in a predetermined

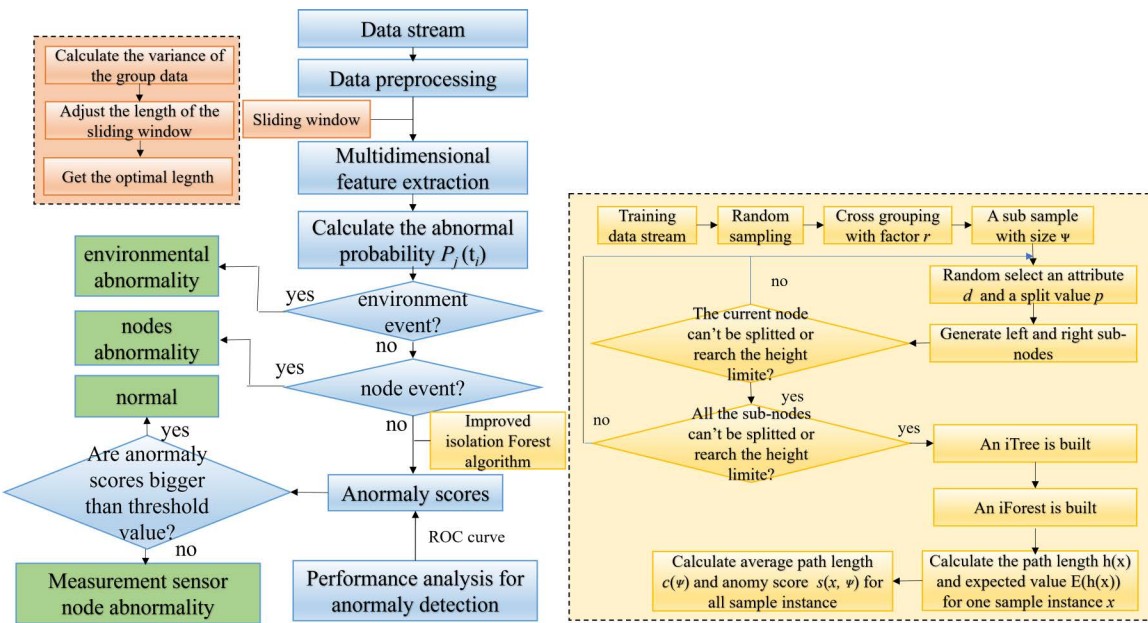

**Fig 1. Flowchart of abnormal detection for CCL.** Firstly, the characteristics of multi-dimensional sensor data are analyzed, the mathematical model of data stream is established, and the sliding window, correlation coefficient and variance are defined. Then, the main three kinds of abnormal events are analyzed, and the judgment conditions of abnormal events based on correlation coefficient are deduced. Finally, a measurement anomaly detection scheme based on the improved iForest algorithm is given, including the algorithm flow chart and the pseudocode of key functions.

order. Data has the characteristics of fast, infinite and continuous arrival. In addition, unlike the general data flow, the sensor data in the CCL has obvious temporal and spatial correlation. First, there is an obvious time correlation between the historical data collected by the same sensor. Then, in order to improve the reliability of the system, multiple sensors of the same type are usually used for multiple azimuth measurements. There is a certain spatial correlation between different sensors. Different types of sensors in the same space also have certain correlation, for example, the temperature and humidity in the space have significant negative correlation. How to make full use of the temporal and spatial correlation between sensor data to improve the accuracy of anomaly detection is one of the issues that need to be considered in sensor data anomaly detection.

Suppose that there are $N$ sensors (such as temperature, humidity, $CO_2$ concentration, etc.) configured to collect $M$ different modes in a mobile terminal for data collection. Each node deployed in wireless sensor network can ensure the synchronization of data acquisition and information transmission through time synchronization mechanism. At a certain sampling time $t$, the data of $M$ different modes collected by any sensor node can be regarded as a set of data points $U = (u_1, u_2,..., u_M)$ in an $M$-dimensional space, and the data collected in a certain sampling period can form a matrix:

$$\{u_j(t_i)\} = \begin{bmatrix} u_1(t_1) & u_1(t_2) & \cdots & u_1(t_N) \\ u_2(t_1) & u_1(t_2) & \cdots & u_1(t_N) \\ \cdots & \cdots & \cdots & \cdots \\ u_M(t_1) & u_1(t_2) & \cdots & u_1(t_N) \end{bmatrix} \tag{1}$$

Where, $t_1$, $t_2$, … $t_N$ is the sample time. Use the sliding window model to process the data stream, as shown in **definition 1**.

**Definition 1:** The Sliding Window Model is to intercept a window with a length of $|W|$ from the sensor data stream, and divide the window into $m$ small blocks, namely $Block_1$, $Block_2$,..., $Block_m$, and the length of each block is $n$. When the data of $t_{next}$ at the next sampling time enters the sliding window, the data of tlast at the last sampling time will be replaced.

$$mod\left(t_{next}, |W|\right) = mod\left(t_{last}, |W|\right) \tag{2}$$

Here, $mod\ (a, b)$ is the remainder function. All sensors on the same sensor node use sliding windows to process data flow internally as **Fig 2**. shown.

Assuming that the data $\{u_j(t_1), u_j(t_2),..., u_j(t_p)\}$ of the first $p$ sampling times of sensor node $j$ are loaded into the sliding window, then the variance of this group of data is

$$\delta^2 = \frac{1}{p} \sum_{i=1}^{p} \prod_{j=1}^{m} \left(u_j(t_i) - \overline{u_j}\right)^2 \tag{3}$$

Here, $\overline{u_j}$ is the average value of the $j$th dimension data collected by the corresponding sensor in the sliding window. When the new data $uj(t_{p+1})$ is loaded into the sliding window, the window slides backward, and the data in the window is updated to $\{u_j(t_2), u_j(t_3),..., u_j(t_{p+1})\}$, and the corresponding calculation formula of sampling data variance can be expressed as:

$$\delta^2 = \frac{1}{p} \sum_{i=2+1}^{p} \prod_{j=1}^{m} \left(u_j(t_i) - \overline{u_j}\right)^2 \tag{4}$$

The data at the subsequent sampling time can be deduced in turn.

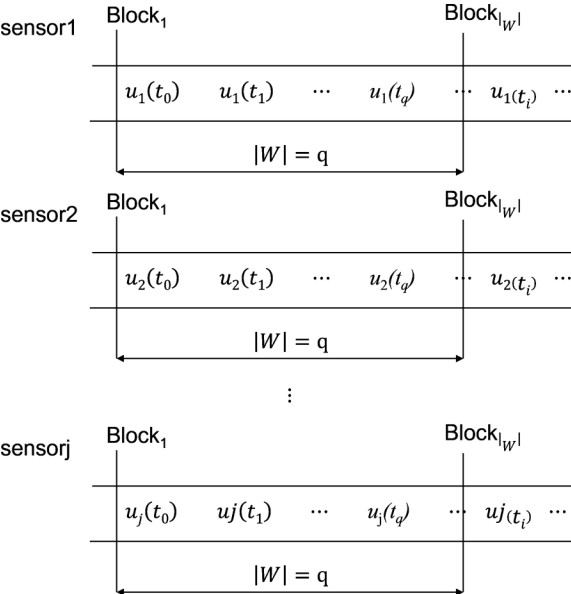

**Fig 2. Sliding window model for multimodal data.**

**Definition 2:** Coherence coefficient $\rho_{ik}$

$$\rho_{ik} = \frac{C(X_i, Y_k)}{\sqrt{C(X_i, Y_i)}\sqrt{C(X_k, Y_k)}} = \frac{\sum_{j=1}^{n}\left(x_{ji} - \overline{X_i}\right)\left(y_{jk} - \overline{Y_i}\right)}{\sqrt{\sum_{j=1}^{n}\left(x_{ji} - \overline{X_i}\right)^2}\sqrt{\sum_{j=1}^{n}\left(y_{jk} - \overline{Y_i}\right)^2}} \tag{5}$$

Here, $x_{ji}$ and $y_{jk}$ represent the $j$th value in the time series of any two data streams $X_i$ and $Y_k$, respectively. The coherence coefficient ρ is an important indicator to evaluate the coherence of multidimensional data flow. If ρ< 0, there is a negative correlation between data streams. If ρ> 0, the data stream is positive coherent. If ρ= 0, data flow has no coherence relationship.

In actual processing, if the sliding window model with width $n$ is used to analyze the coherence of multiple data streams, the covariance matrix S11 of data streams $X_i$ = ($X_{1i}$, $X_{2i}$,..., $X_{pi}$) needs to be calculated, the covariance matrix S22 of data streams Yi = ($Y_{1i}$, $Y_{2i}$,..., $Y_{pi}$) needs to be calculated, the covariance matrix S12 of data streams $X_i$ and $Y_{i\cdot}$ as shown in formula 6. After standardizing the data, the correlation coefficient of the sample corresponds to the covariance of the sample. Finally, the corresponding typical correlation coefficient and typical correlation variable can be obtained by using the chi-square test method. For example, when a fire occurs in a CCL carriage, the sensor nodes collect temperature values that have a significant positive correlation with $CO_2$ concentration values and a negative correlation with humidity values. Therefore, our research on the coherence and spatiotemporal correlation between multidimensional data in wireless sensor networks can provide a theoretical basis for accurate and efficient anomaly detection. On this basis, this article proposes a method for anomaly detection of WSN multimodal data streams.

$$S_{11} = \frac{1}{n-1}\sum_{i=1}^{n}\left(X_i - \overline{X}\right)\left(X_i - \overline{X}\right)^T$$

$$S_{22} = \frac{1}{n-1}\sum_{i=1}^{n}\left(Y_i - \bar{Y}\right)\left(Y_i - \bar{Y}\right)^T$$

$$S_{12} = \frac{1}{n-1}\sum_{i=1}^{n}\left(X_i - \bar{X}\right)\left(Y_i - \bar{Y}\right)^T = S_{21}^T \tag{6}$$

## Classification and judgment of main abnormalities in CCL

The main reasons for the abnormal data generated by the CCL mobile terminal include: (1) Some specific events occurs in the mobile terminal, for example, in case of water leakage in the movie carriage, the temperature reading of the sensor will decrease significantly, which is called an environment event; (2) When the node's circuit cannot work normally, all parameters reading data of the node will abnormal, which is called a node event; (3) The data collected by the node deviates from the normal data because of the influence of external factors, which is called a measurement event. Abnormal data derived from specific events often reflect that some significant changes have occurred, which needs to be dealt immediately. However, the abnormal data caused by sensor node, which needs to be maintained. Because the data from the measurement anomaly cannot represent the actual environmental characteristics, in order to make accurate judgments, it is necessary to detect the data collected by the wireless sensor network in order to find the abnormal data in time and analyze and identify its source.

The measured value of the sensor should accurately reproduce the actual environmental characteristics, so the measured value $u_j(t_i)$ should fluctuate slowly within a certain range in a stable environment, and when there is an abnormality, there will be a significant deviation in a short time. If $u_j(t_i)$ meets equation (7), then the measured value may be abnormal data.

$$\left| u_j\left(t_i\right) - \frac{E_{ej}\left(t\right) + E_{nj}\left(t\right)}{2} \right| > \delta^2 \tag{7}$$

Here, $t$ is the sample time. $E_{ej}(t)$ is the mathematical expectation of the measured value of the normal working sensor in the event area. $E_{nj}(t)$ is the mathematical expectation of the measured value in the normal area. It is generally considered that $E_{nj}(t)$ is a constant under stable conditions. The values of $E_{ej}(t)$ and $E_{nj}(t)$ in different environments are different, which is determined by the data set.

When the sensor fails (energy is exhausted or damaged and cannot work normally), the same data may be continuously generated at different sampling times, that is

$$u_j\left(t_i\right) = u_j\left(t_{i-1}\right) \tag{8}$$

The above two cases are called the judgment conditions to determine whether the sensor data is abnormal, and calculate the abnormal probability $P_j(t_i)$ of single-mode data flow based on this.

$$P_j\left(t_i\right) = P_j\left(t_{i-1}\right) + c \cdot k^2 \tag{9}$$

Here, constant $k$ represents the number of times $\{u_j(t_i)\}$ satisfies the judgment condition. $c$ is a parameter. If $u_j(t_i)$ continuously meets the judgment conditions at several sampling times, then $k$ will increase gradually from 0. At this time, $P_j(t_i)$ and $k$ are exponential. If $u_j(t_i)$ does

not meet the judgment condition, $k$, $P_j(t_{i-1})$ and $P_j(t_i)$ are cleared at the same time. When $u_j(t_i)$ meets the judgment condition, $k$ starts to accumulate again.

The sensor node has a variety of sensors, so at a certain sampling time, the sensor node has multiple modes of data flow, generating multiple sets of $P_j(t_i)$. It is not accurate to judge the cause of data anomaly only through a single mode data flow. It needs to fuse multi-mode data flow for analysis and judgment. The multi-mode exception probability $P_T(t_i)$ can be calculated from the multi-group single-mode exception probability $P_j(t_i)$ value as follows:

$$P_T\left(t_i\right) = \sum_{j=1}^{m} \lambda_j \cdot P_j(t_i), \sum_{j=1}^{m} \lambda_j = 1 \tag{10}$$

Where $\lambda_j$ is the weight coefficient. In consideration of $\lambda_j$ is related to the fluctuation range of the data, which can be consistent with the standard deviation ratio of the data in proportion, that is

$$\lambda_1 : \lambda_2 : \cdots : \lambda_j = \sigma_1 : \sigma_2 : \cdots : \sigma_j \tag{11}$$

When the PT(ti) of a sensor node reaches the threshold $R_{th} = \sum_{m}^{j=1} \lambda_j \cdot \dfrac{E_{ej}\left(t\right) + E_{nj}\left(t\right)}{2}$, it is

considered that the node may have an exception. Here, Rth is set as the weighted average of the mean of the multimodal data set. Next, we need to use the spatial correlation of nodes to determine the type of exception. The sensor should receive the $P_T(t_i)$ of the neighbor node. According to the Pauta criterion, if the $P_T(t_i)$ of the node meets $\left|P_T\left(t_i\right) - \mu\right| < \delta\sigma$ ($\mu$ and $\sigma$ are mean and standard deviation of $P_T(t_i)$ of neighbor node respectively), then it is considered that the error comes from the random error in the event process, and the state of the node is consistent with that of neighbor node; If not, it is considered that the status of this node is inconsistent with that of the adjacent node, and there is a fault node event or measurement event. The value of $\delta$ depends on the specific situation. Generally, the event can be regarded as a Bernoulli process with normal distribution of random variables, so the random variable can be simplified as a random variable with standard normal distribution.

$$p = P\left(\frac{\left|P_T(t_i) - \mu\right|}{\sigma} \geq \delta\right)$$

$$= 1 - P\left(-\delta < \frac{P_T(t_i)}{\sigma} < \delta\right)$$

$$= 2 - 2\Phi\left(\delta\right) \tag{12}$$

Here, $\Phi(\delta)$ is the standard normal distribution function. It can be seen from the table when $\Phi(\delta) > 0.975$, $p < 0.05$. when $\delta$ is more than 1.96, $\Phi(\delta) > 0.975$, so we can take $\delta=2$.

Then, three kinds of the abnormal event can be judged as follows:

If $u_j(t_i) \neq u_j(t_{i-1})$ and $\left|P_T\left(t_i\right) - \mu\right| < 2\sigma$, it can be considered an environment event;

If $u_j(t_i) = u_j(t_{i-1})$ and $\left|P_T\left(t_i\right) - \mu\right| \geq 2\sigma$, it can be considered a node event;

If the above conditions are not met, it can be considered as a measurement event. Next, we need to detect and filter the measured data. In the actual CCL system, the measurement event is the most important exception among three abnormalities.

## Anomaly detection based on improved iForest algorithm

The node with measurement error has first eliminated the possibility of node failure event and environment abnormal event, and should be a node that can work normally. However, there

are data in the collected data stream that are significantly different from the actual environmental characteristics, so it is necessary to detect the nodes with measurement errors and find out the abnormal data, so as to improve the reliability of the CCL system.

Isolation forest algorithm is widely used in anomaly detection, because its computing speed is fast, unlike clustering-based algorithms such as *K*-means, which spend a lot of time computing distance, and its robustness is strong. The steps of the algorithm are as follows:(i) People randomly select a part of the samples from all the data as a set of isolated trees, randomly select one dimension and one segmentation point for segmentation, and divide the data into two subspaces in this dimension, that is, less than the segmentation point or greater than or equal to the segmentation point. (ii) People continuously select dimensions and segmentation points randomly, and operate repeatedly until there is only one sample data in a certain subspace, or all attribute values in the subspace are the same (cannot be segmented), or the height of the previously preset tree has been reached. When any one of the above three conditions is met, the construction of the orphan tree is stopped. (iii) The above *N* trees are built to form an isolated forest. (iv) The sample score is constructed, that is, after the final isolated forest is formed, the position of the tree where each sample is located is scored, and the total score is obtained. The closer the score is to 1, the more likely it is to be abnormal. The closer it is to 0, the more likely it is to be normal. When the score is close to 0.5, it is impossible to judge whether it is abnormal or normal. The schematic diagram of the algorithm is shown in **Fig 3**.

However, the disadvantage of this algorithm is also obvious, and it is easy to fall into local optimum, which is attributed to randomly selecting arbitrary one-dimensional data for cutting. To overcome this shortcoming, this paper proposes an improved isolated forest algorithm based on full feature fusion.

In the process of building the traditional isolated forest model, because each time the data space is cut into subtrees, a feature is randomly selected from all features, so more feature information may not be used after all features are built, resulting in low accuracy of the algorithm. To solve this problem, the feature set of the data set is cross grouped during the process of building the isolated forest model. Define $Q$ as the all feature set of the data set, and $n$ empty feature subset $Q_i$ ($i = 1,2... n$) of $Q$. The number of elements in feature set $Q$ is $m$. The number of elements in the feature subset $Q_i$ is $l$. $r$ is defined as the cross factor, which represents the proportion of the number of features contained in each feature subset in the number of all features in the data set, that is, $r = l/m$, with a value of $0 \sim 1.0$. Then put each feature in the feature set into the feature subset in turn, such as putting the first feature in $Q$ into $Q_1,… Q_m$, the second feature into $Q_2...Q_{m+1}$, the $i$-th feature into $Q_{(i-1)mod\,n+1}...Q_{(i+m-2)mod\,n+1}$, and the $i + 1$ feature into $Q_{i\,mod\,n+1}...Q_{(i+m-1)mod\,n+1}$. A certain number of isolated trees are constructed on each feature subset after cross grouping. The model construction process is as follows:

Step 1: Input the feature data set and initialize the parameters of the iForest for balanced modeling of full feature information. Here, set the number of iTrees to 100, the sub-sample size to 256, and the tree height to 8.

Step 2: The sub-sampling algorithm is used to realize the sub-sampling of the data set, and the set composed of all features of the data set after sub-sampling is divided according to the cross-grouping rules. Let all feature sets of the dataset be $Q$, and the number of elements in the set be $m$. After cross grouping, multiple feature subsets are generated, which are $Q_1, Q_2, …, Q_n$. The number of elements in each subset is $l$, and all feature subsets should meet the condition $Q_1 \cup Q_2 \cup \cdots \cup Q_n = Q$, cross factor $r = l/m$, adjusted by fixed $n$ according to the test effect of verification set.

Step 3: Build iTree using the above sample space. During the construction process, each time the data space is cut for subtree division, the feature values are selected from the feature set elements in step 2. After determining the selected feature, find the value range of the current feature, and randomly select a split value within the range to complete the subtree

division of the data space. Repeat the process until the termination condition is reached, and complete the construction of the isolation tree.

The pseudocode of the iTree is summarized in the **Table 1**.

Isolated forests is built based on isolated trees. To ensure differences between different trees, a method of randomly sampling partial datasets is used to construct each isolated tree. A certain number of isolated trees are built for each feature subset during the process of building isolated forests. Finally, multiple sets of isolated trees generated based on different feature subsets will be integrated together to form an isolated forest. The pseudocode of the isolated forest is summarized in the **Table 2**.

Isolated forests is built based on isolated trees. To ensure differences between different trees, a method of randomly sampling partial datasets is used to construct each isolated tree. A certain number of isolated trees are built for each feature subset during the process of building

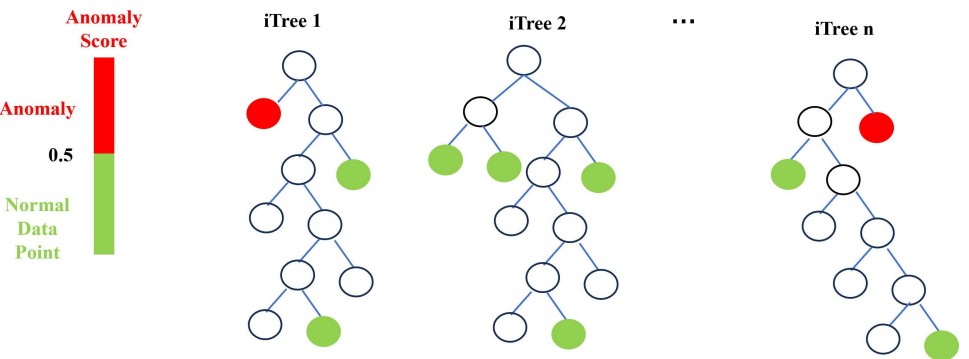

**Fig 3. The schematic diagram of the iForest algorithm.**

**Table 1. The pseudocode of the iTree.**

| Algorithm 1: isolatedTree(*X'*, *Qn*) |
|---|
| **Input**:                input sub-sampling data $X'$; the Group of feature $Q_n$; |
| **Output**:        an iTree; |
| **Procedure**: |
| 1: **begin** |
| 2: **if** $X'$ Can't be divided **then** |
| 3: return iTree; |
| 4: **else** |
| 5: Randomly select a feature $q$ from $Q_n$, $q \in Q_n$; |
| 6: **if** $q$==$Last_q$ **then** |
| 7: $q$ = neighbor($q$); |
| 8: Randomly select a value $Value_q$ from $X'_q$; |
| 9: $X_l$ = Filter($X'$,$X'_q \leq Value_q$); |
| 10: $X_r$ = Filter($X'$,$X'_q > Value_q$); |
| 11: Return inNode{ |
| 12: Left = Itree($X_l$); |
| 13: Right = Itree($X_r$); |
| 14: SplitAtt = $q$; |
| 15: SplitValue = $X'_q$;} |
| 16: **end if** |
| 17: **end** |

**Table 2. The pseudocode of the isolated Forests.**

| Algorithm 2: IsolationForest (*Data,Numtree*, *NumSmp,n,r*) |
|---|
| **Input**:          Input sampling *Data*; the num of IsolatoinTrees *NumTree*; |
|          Data sampling band *NumSmp*; Feature group num *n*; crossover factor *r*; |
| **Output**:          Forest—Group of all Itrees |
| **Procedure**: |
|          1: **begin** |
|          2:          initialize Forest |
|          3:   **for** *i* = 1; *i* ≤ *Data.featurenum*; **do** |
|          4:          Send i-th feature to Q$_{(i-1)mod\,n+1}$,…,Q$_{(i+rn-2)mod\,n+1}$ |
|          5: **end for** |
|          6: **for** *j* = 1; *i* ≤ NumTree; **do** |
|          7:          *X'* = Sample(Data,NumSmp) |
|          8:          Forest = Forest ∪ Itree(*X'*,Q$_{mod(i,n)+1}$) |
|          9: **end for** |
|          10: **end** |

isolated forests. Finally, multiple sets of isolated trees generated based on different feature subsets will be integrated together to form an isolated forest.

The final judgment is determined by the abnormal score. The abnormal score is defined by expression (13).

$$s(x,n) = 2^{(-\frac{E(h(x))}{c(n)})} \qquad (13)$$

The value of *s(x,n)* is [0, 1.0]. The closer the value of *s(x,n)* is to 1.0, the more isolated the sample is, and the more possible it is to be abnormal. The average path evaluation method of sorted binary tree is used to normalize the results. The *c(n)* is defined as follows:

$$c(n) = 2H(n-1) - 2(n-1)/n \qquad (14)$$

here, $H(k) = \ln k + \xi$, $\xi$ is Euler constant, it is 0.5772156649. *h(x)* is the path length of the test data traversing a single isolated tree, and *E(h(x))* is the average path length of the test data traversing each isolated tree. The pseudocode of the length calculation is summarized in the **Table 3**.

In the construction process of isolated tree, the selection range of features is all elements in the subset, which effectively improves the balance of the use of feature information in the construction process of isolated tree and realizes the full use of data feature information. This method not only overcomes the shortcomings of high dimension, but also does not increase the workload of the algorithm. It only needs to try the value of the cross factor. It is a fast and efficient improved isolated forest algorithm.

## Experiment and results

This section describes the experiments. First, a brief description of experimental data and environment is given. Then four experimental results are given, including the parameter selection of the optimal sliding window length and the optimal cross factor, the performance comparison about improved iForest and iForest in *P*, *R*, *F1* score and AUC, the performance comparison about improved iForest, iForest, SVM and LOF in AUC and running time.

### Data and environment

The data collection equipments were placed on three CCL vehicles of a CCL company in Ningbo, Zhejiang Province, China, to collect real-time sensor data from the mobile vehicles and conduct

**Table 3. The pseudocode of the length calculation.**

| Algorithm 3: IsolationMass (*TestData,Currentheight, Limitheight,Itree*) |
|---|
| **Input:** Input test data *TestData*; current tree height *Currentheight*;<br> tree limit height *Limitheight*; Isolation tree *Itree*;<br>**Output:** the current data's score in this Itree.<br>*c = 2·(log(Tree.size-1) + 0.5772156649)-2(Tree.size-1)/Tree.size*<br>**Procedure:**<br> **1: begin**<br> **2:** **if** *Itree* is external node or *Currentheight ≥ Limitheight* **then**<br> **3:** **return** $e^+c$(*Itree.size*)<br> **4:** **end if**<br> **5: if** $TestData_{splitAtt}$ ≤ *Itree.splitvalue*<br> **6:** **return** IsolationMass(*TestData,Currentheight*$^+$1,*Limitheight,Itree.left*)<br> **7:** **else** $TestData_{splitAtt}$ > *Itree.splitvalue*<br> **8:** **return** IsolationMass(*TestData,Currentheight*$^+$1,*Limitheight,Itree.right*)<br> **9:** **end if**<br> **10: end** |

experiments from December 1st to December 7th, 2023. There are 30 nodes in each compartment, and each node has 5 sensors to detect the temperature, humidity, concentration of $CO_2$, concentration of $O_2$ and pressure near the node as shown in **Table 4**. Therefore, there are 150 dimensional data streams in each carriage. Each sensor node sends detection parameters to the sink node every minute. So, the experimental dataset is a 150 dimensional real-time data stream with characteristic parameters including temperature, humidity, oxygen concentration, carbon dioxide concentration, and pressure values. The data for each feature parameter is 40 bits. So the dataset in each carriage is 30 Mbits. The experiment was carried out in a Lenovo desktop computer with Windows 11 operating system, which was configured with Intel Core i7 CPU and 16 GB memory.

## Performance indicators

Precision *P*, recall *R*, *F1* score, AUC and algorithm executing time are selected as the indicators to evaluate the performance. Confusion matrix is shown as **Table 5**. The purpose of the confusion matrix is to compare actual and predicted labels. The terms TP (true positive) and TN (true negative) denote correctly predicted conditions and FP (false positive) and FN (false negative) misclassified ones. TPs and TNs refer to correctly classified abnormal and normal records, respectively and, conversely, FPs and FNs refer to misclassified normal and abnormal records, respectively.

Based on the confusion matrix model, the *P* represents the proportion of samples that are truly positive in the samples that are predicted to be positive. The greater the accuracy, the better the prediction performance. *P* is defined as follows:

$$P = TP / (TP + FP) \tag{15}$$

*R* refers to the ratio of the number of predicted positive samples to the total number of positive samples. The higher the *R* is, the more positive samples are predicted correctly and the better the prediction performance is. *R* is defined as follows:

$$R = TP / (TP + FN) \tag{16}$$

*F1* score is a harmonious mean of *P* and *R*, that is, a statistical function for estimating the accuracy of a system by computing its *P* and *R* given as:

$$F1 = 2PR / (P + R) \tag{17}$$

**Table 4. Vehicle information and parameters in the experiment.**

| Vehicle type | Goods | Volume (m³) | Temperature (°C) | Number of nodes | Number of instances | Detection parameters |
|---|---|---|---|---|---|---|
| A | fruits | 15 | 2 ~ 8 | 30 | 5K | temperature |
| D | eggs, meat, | 17 | 0 ~ -5 | 30 | 5K | $CO_2$ |
| F | sea foods | 15 | ≤-18 | 30 | 5K | $O_2$ humidity pressure |

**Table 5. Confusion matrix for calculating the abnormal detection.**

| Sample category | Actual negative | Actual positive |
|---|---|---|
| Predicted negative | TN | FP |
| Predicted positive | FN | TP |

ROC (receiver operating characteristic curve) was created from the signal processing theory and then extended to other domains, such as data mining and machine learning as well as artificial intelligence. ROC curve is based on a series of different binary classification methods. FPR is the false positive rate, TPR is the true rate, and the area covered by the curve is defined as AUC. ROC curve and AUC are commonly used to evaluate the advantages and disadvantages of classification models, and the larger the AUC, the better the prediction effect.

The running time of the algorithm is one of the important indicators to measure the performance of the algorithm. The running time of the algorithm includes model training time and model verification time.

## Results and analysis

First, we need to determine the most optimal legnth of the sliding window. In terms of accuracy, the larger the window length, the higher the accuracy. However, the larger the window length is, the larger the calculation amount is. In order to investigate the impact of sliding windows of different lengths on the statistical characteristics of data flow, 10000 data groups of one-dimensional temperature, $CO_2$ concentration, $O_2$ concentration, humidity and pressure were selected and calculated with the square difference of sliding windows of different lengths. The results are shown in **Table 6**. The variance of data flow tends to be stable with the increase of window length. It can be seen from **Table 6** that when the length of the sliding window is greater than 200, the variance tends to be stable. Therefore, the best length of the sliding window here is 200.

Then we study the optimal value of the cross factor $r$. According to reference [1], number of iTree is set to 100, number of sub samples is set to 256, number of cross groups is set to 10, cross factor $r$ varies from 0.1 to 1.0, and step length of cross factor is set to 0.1. The relationship between $P$, $R$, $F1$ score, AUC and cross factor $r$ are shown in **Fig 4**. The improved iForest algorithm is better than the original algorithm in $P$, $R$, $F1$ score and AUC when $r$ is 0.5 ~ 0.9. When $r$ is 0.6, the above performance is optimal. In this experiment, different cross factors determine the number of features in the feature subset of the isolated tree. When $r$ is less than 0.4, the number of elements in the grouped feature subset is too small, resulting in insufficient information contained in the isolated tree and poor detection effect. When $r$ is 0.5 ~ 0.9, the isolated tree constructed based on the grouped feature subset realizes the balanced utilization of feature information, and the detection effect is improved. From the line graph, it can be seen that when $r$ is set to 0.6, the detection effect is the best; When $r$ is 1.0, each isolated tree is constructed using all features. So the cross factor is set to 0.6 in the later experiments.

**Table 6. Variance of data stream of different sliding window length.**

| Length of window | Temperature variance | $CO_2$ concentration variance | $O_2$ concentration variance | Humidity variance | Pressure variance |
|---|---|---|---|---|---|
| 10 | 6.12 | 5.54 | 3.57 | 6.14 | 4.97 |
| 50 | 5.83 | 5.25 | 3.45 | 6.03 | 4.90 |
| 100 | 5.75 | 4.92 | 2.89 | 5.94 | 4.86 |
| 150 | 5.59 | 4.79 | 2.60 | 5.82 | 4.80 |
| 180 | 4.84 | 4.31 | 2.37 | 5.45 | 4.75 |
| 200 | 4.38 | 4.17 | 2.19 | 5.37 | 4.70 |
| 220 | 4.36 | 4.17 | 2.18 | 5.35 | 4.69 |
| 250 | 4.37 | 4.16 | 2.17 | 5.34 | 4.67 |
| 280 | 4.36 | 4.16 | 2.19 | 5.34 | 4.67 |
| 300 | 4.38 | 4.17 | 2.17 | 5.33 | 4.66 |
| 400 | 4.37 | 4.18 | 2.18 | 5.34 | 4.66 |

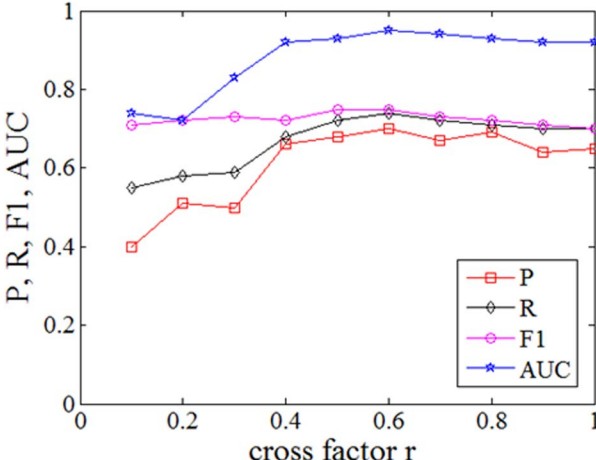

**Fig 4. The performance indicators P, R, F1 score, AUC under different cross factor r.**

Thirdly, we take the comparison experiments about main algorithms currently used. In addition to the iForest algorithm, SVM and LOF are also the most commonly used anomaly detection algorithms. We tested the performance of the four algorithms with the three vehicle A, D and F in a local CCL company. There are 30 nodes in a van. We selected 5, 10, 15, 20, 25 and 30 nodes for experiments to verify the impact of different dimensional data streams on the experimental results. **Table 7** shows the performance indicators with *P, R, F1* score and AUC of each vehicle with 5 nodes to collecting the sensors data streams. Obviously, the proposed algorithm do not have particularly obvious advantages.

**Table 8** shows the performance indicators with *P, R, F1,* and AUC of each vehicle with 10 nodes. It is clear that the iForest algorithm and the improved iForest algorithm have some advantage in performance indicators when the dimensionality of the data stream increasing.

**Table 9** shows the performance indicators with *P, R, F1,* and AUC of each vehicle with 15 nodes to collecting the sensors data streams. It is clear the improved iForest algorithm have obvious advantage in performance indicators. And the performance indicators of the improved iForest algorithm are better than those of the iForest algorithm.

**Table 7. The comparison of performance indicators of four algorithms with 5 nodes.**

| Vehicle type | Algorithm | Performance indicators | | | |
|---|---|---|---|---|---|
| | | P | R | F1 | AUC |
| A | LOF | 0.6317 | 0.6275 | 0.7038 | 0.6694 |
| | SVM | 0.5871 | 0.7139 | 0.6327 | 0.7236 |
| | iForest | 0.5310 | 0.7517 | 0.6749 | 0.6549 |
| | proposed | 0.5293 | 0.7583 | 0.6649 | 0.6273 |
| D | LOF | 0.6109 | 0.7386 | 0.6483 | 0.6937 |
| | SVM | 0.6275 | 0.7471 | 0.6041 | 0.7729 |
| | iForest | 0.6437 | 0.7729 | 0.6965 | 0.6761 |
| | proposed | 0.6105 | 0.7326 | 0.6738 | 0.6559 |
| F | LOF | 0.5832 | 0.6962 | 0.6371 | 0.6796 |
| | SVM | 0.5618 | 0.6759 | 0.5961 | 0.7275 |
| | iForest | 0.6196 | 0.8475 | 0.7572 | 0.6193 |
| | proposed | 0.5938 | 0.7404 | 0.6503 | 0.6395 |
| Average | LOF | 0.6086 | 0.6874 | 0.6630 | 0.6809 |
| | SVM | 0.5921 | 0.7123 | 0.6109 | 0.7413 |
| | iForest | 0.5981 | 0.7907 | 0.7095 | 0.6501 |
| | proposed | 0.5778 | 0.7437 | 0.6630 | 0.6409 |

**Table 8. The comparison of performance indicators of four algorithms with 10 nodes.**

| Vehicle type | Algorithm | Performance indicators | | | |
|---|---|---|---|---|---|
| | | P | R | F1 | AUC |
| A | LOF | 0.6631 | 0.7127 | 0.6383 | 0.7149 |
| | SVM | 0.5923 | 0.7013 | 0.6732 | 0.7217 |
| | iForest | 0.5827 | 0.8109 | 0.8419 | 0.8507 |
| | proposed | 0.6293 | 0.8237 | 0.8440 | 0.8573 |
| D | LOF | 0.6610 | 0.7024 | 0.6481 | 0.7432 |
| | SVM | 0.6301 | 0.7382 | 0.6091 | 0.7491 |
| | iForest | 0.6735 | 0.8220 | 0.7995 | 0.8163 |
| | proposed | 0.6105 | 0.8339 | 0.8512 | 0.8552 |
| F | LOF | 0.6529 | 0.7517 | 0.6373 | 0.7157 |
| | SVM | 0.5305 | 0.7024 | 0.5983 | 0.7198 |
| | iForest | 0.6387 | 0.7890 | 0.7919 | 0.7937 |
| | proposed | 0.6938 | 0.8416 | 0.8374 | 0.8374 |
| Average | LOF | 0.6590 | 0.7223 | 0.6412 | 0.7246 |
| | SVM | 0.5843 | 0.7139 | 0.6268 | 0.7302 |
| | iForest | 0.6316 | 0.8073 | 0.8111 | 0.8202 |
| | proposed | 0.6445 | 0.8330 | 0.8442 | 0.8450 |

Table 10 is shown the performance indicators with *P, R, F1,* and AUC of each vehicle with 20 nodes. It is clear that the proposed algorithm have obvious advantage in performance indicators among all algorithms.

Table 11 and Table 12 show the performance indicators with *P, R, F1,* and AUC of each vehicle with 25 and 30 nodes to collecting the sensors data streams. It is obvious that the performance indicators of the proposed algorithm are better than other algorithms, because as

**Table 9. The comparison of performance indicators of four algorithms with 15 nodes.**

| Vehicle type | Algorithm | Performance indicators | | | |
|---|---|---|---|---|---|
| | | P | R | F1 | AUC |
| A | LOF | 0.6316 | 0.6275 | 0.7038 | 0.6694 |
| | SVM | 0.5868 | 0.7139 | 0.6327 | 0.7236 |
| | iForest | 0.7301 | 0.7517 | 0.7749 | 0.8049 |
| | proposed | 0.8292 | 0.8583 | 0.8649 | 0.8273 |
| D | LOF | 0.6117 | 0.7386 | 0.6483 | 0.6937 |
| | SVM | 0.6257 | 0.7471 | 0.6041 | 0.7729 |
| | iForest | 0.8473 | 0.7729 | 0.7965 | 0.8261 |
| | proposed | 0.8506 | 0.8326 | 0.8738 | 0.8559 |
| F | LOF | 0.5836 | 0.6962 | 0.6371 | 0.6796 |
| | SVM | 0.5617 | 0.6759 | 0.5961 | 0.7275 |
| | iForest | 0.8194 | 0.8475 | 0.8572 | 0.7693 |
| | proposed | 0.7535 | 0.8404 | 0.8503 | 0.8395 |
| Average | LOF | 0.6089 | 0.6874 | 0.6630 | 0.6809 |
| | SVM | 0.5914 | 0.7123 | 0.6109 | 0.7413 |
| | iForest | 0.7989 | 0.7907 | 0.8095 | 0.8001 |
| | proposed | 0.8111 | 0.8437 | 0.8630 | 0.8409 |

**Table 10. The comparison of performance indicators of four algorithms with 20 nodes.**

| Vehicle type | Algorithm | Performance indicators | | | |
|---|---|---|---|---|---|
| | | P | R | F1 | AUC |
| A | LOF | 0.5815 | 0.6257 | 0.7053 | 0.6693 |
| | SVM | 0.5872 | 0.7138 | 0.6328 | 0.7234 |
| | iForest | 0.7309 | 0.7571 | 0.7747 | 0.8547 |
| | proposed | 0.8295 | 0.8585 | 0.8648 | 0.8871 |
| D | LOF | 0.5610 | 0.7384 | 0.6485 | 0.6939 |
| | SVM | 0.6273 | 0.7470 | 0.6045 | 0.7731 |
| | iForest | 0.8433 | 0.7723 | 0.7969 | 0.8764 |
| | proposed | 0.9107 | 0.8362 | 0.8739 | 0.9161 |
| F | LOF | 0.5334 | 0.6963 | 0.6373 | 0.6793 |
| | SVM | 0.5617 | 0.6757 | 0.5965 | 0.7274 |
| | iForest | 0.8198 | 0.8474 | 0.8574 | 0.8190 |
| | proposed | 0.8983 | 0.8409 | 0.8530 | 0.8993 |
| Average | LOF | 0.5586 | 0.6868 | 0.6111 | 0.6808 |
| | SVM | 0.5920 | 0.7122 | 0.6113 | 0.7413 |
| | iForest | 0.7980 | 0.7923 | 0.8097 | 0.8500 |
| | proposed | 0.8795 | 0.8452 | 0.8639 | 0.9008 |

the dimension increases, the computation of SVM and LOF algorithms will sharply increase, and iForest algorithm is prone to falling into local optima.

The average performance indicators of different nodes under four algorithms are shown in **Fig 5**. When the dimensionality is low, the advantages of the proposed algorithm are not significant, while as the dimensionality increases, the performance indicators of the proposed algorithm are significantly higher than the other three algorithms.

**Table 11. The comparison of performance indicators of four algorithms with 25 nodes.**

| Vehicle type | Algorithm | Performance indicators | | | |
| --- | --- | --- | --- | --- | --- |
| | | P | R | F1 | AUC |
| A | LOF | 0.5361 | 0.5273 | 0.6037 | 0.6692 |
| | SVM | 0.5570 | 0.6137 | 0.5726 | 0.6732 |
| | iForest | 0.7909 | 0.7816 | 0.7748 | 0.8248 |
| | proposed | 0.8293 | 0.8983 | 0.8648 | 0.8974 |
| D | LOF | 0.5108 | 0.6388 | 0.5484 | 0.6932 |
| | SVM | 0.5973 | 0.7447 | 0.5442 | 0.6727 |
| | iForest | 0.8238 | 0.8030 | 0.7967 | 0.8562 |
| | proposed | 0.9105 | 0.8826 | 0.8739 | 0.9258 |
| F | LOF | 0.4823 | 0.5932 | 0.5373 | 0.6805 |
| | SVM | 0.5316 | 0.5760 | 0.5363 | 0.7274 |
| | iForest | 0.7895 | 0.8772 | 0.8571 | 0.7895 |
| | proposed | 0.8938 | 0.8904 | 0.8505 | 0.9095 |
| Average | LOF | 0.5097 | 0.5864 | 0.5631 | 0.6810 |
| | SVM | 0.5619 | 0.6448 | 0.5510 | 0.6911 |
| | iForest | 0.8014 | 0.8206 | 0.8095 | 0.8235 |
| | proposed | 0.8779 | 0.8904 | 0.8649 | 0.9109 |

**Table 12. The comparison of performance indicators of four algorithms with 30 nodes.**

| Vehicle type | Algorithm | Performance indicators | | | |
| --- | --- | --- | --- | --- | --- |
| | | P | R | F1 | AUC |
| A | LOF | 0.5015 | 0.5257 | 0.5437 | 0.5649 |
| | SVM | 0.5417 | 0.6140 | 0.5626 | 0.6234 |
| | iForest | 0.7311 | 0.7519 | 0.7550 | 0.8104 |
| | proposed | 0.8294 | 0.8936 | 0.8650 | 0.9270 |
| D | LOF | 0.4811 | 0.6368 | 0.4884 | 0.5973 |
| | SVM | 0.5885 | 0.6471 | 0.5343 | 0.6727 |
| | iForest | 0.8436 | 0.7726 | 0.7764 | 0.8392 |
| | proposed | 0.9103 | 0.8762 | 0.8736 | 0.9058 |
| F | LOF | 0.4533 | 0.5982 | 0.4772 | 0.5797 |
| | SVM | 0.5281 | 0.5760 | 0.5260 | 0.6280 |
| | iForest | 0.8195 | 0.8474 | 0.8371 | 0.7693 |
| | proposed | 0.8940 | 0.8809 | 0.8504 | 0.8896 |
| Average | LOF | 0.4786 | 0.5869 | 0.5031 | 0.5806 |
| | SVM | 0.5528 | 0.6123 | 0.5410 | 0.6414 |
| | iForest | 0.7981 | 0.7906 | 0.7895 | 0.8063 |
| | proposed | 0.8779 | 0.8836 | 0.8630 | 0.9075 |

It is obvious that the iForest algorithm and the proposed algorithm in this paper do not have significant advantages, when the dimensionality of the data stream is low, such as when the number of nodes is 5, and some performance indicators are not as good as those of LOF and SVM algorithms. When the dimensionality of the data stream increases, the iForest algorithm and the proposed scheme have significant advantages in performance metrics, such as when the number of nodes is greater than 10. The performance of iForest and the proposed

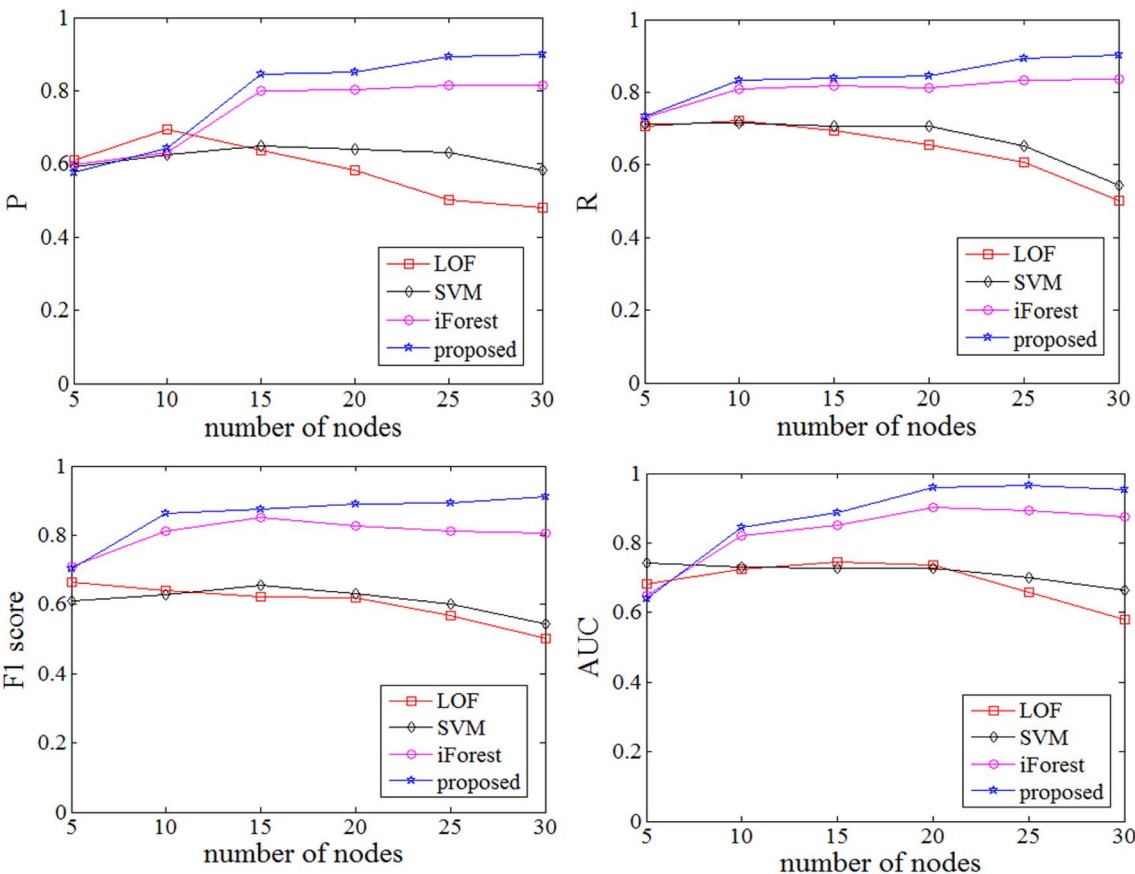

**Fig 5. Comparison of performance indicators with different node number.**

algorithm are very similar when data dimensions are low. When the data dimension is large, the performance indicators of the proposed scheme is better than that of iForest, because it can overcome the disadvantage of iForest algorithm being easily limited to local optima in high dimensions. When the number of nodes is 30, the $P$ is 0.8779, $R$ is 0.8836, $F1$ score is 0.8630, AUC is 0.9075, which is more better than those of other three algorithms.

The ROC curves of the four algorithms with 30 and 5 nodes are compared in **Fig 6** and **Fig 7**. Similar to the four performance indicators mentioned above, when the data dimensionality is low, the ROC of the proposed algorithm is not significantly better than the other three algorithms, while when the data dimensionality is high, its ROC is significantly better than the other three algorithms.

The executing time comparison of the four algorithms is shown in **Table 13**. The executing time of the improved algorithm is a little more than that of the iForest, but it is significantly less than that of the SVM algorithm and LOF algorithm. This is because the computation amount of improved algorithm and iForest is O $(n)$, while that of SVM and LOF algorithm is O $(n^2)$.

## Discussion

Through the experiments above, three issues should be discussed.

First of all is the slide window length. The optimal length 200 of the sliding window for multidimensional data streams is determined in experiments based on actual measurements. The value should be different in different experimental environments. Although theoretically,

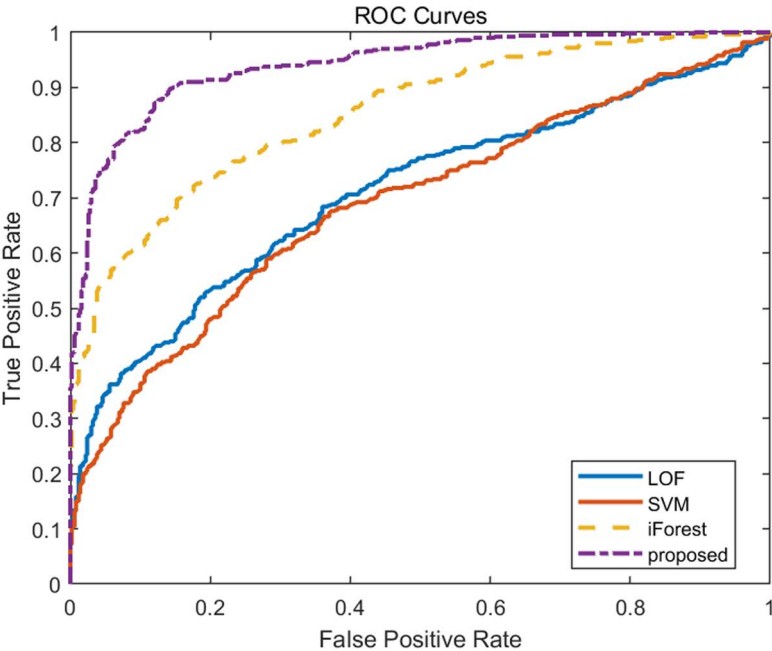

**Fig 6. Comparison of ROC curves for four algorithms with 30 nodes.**

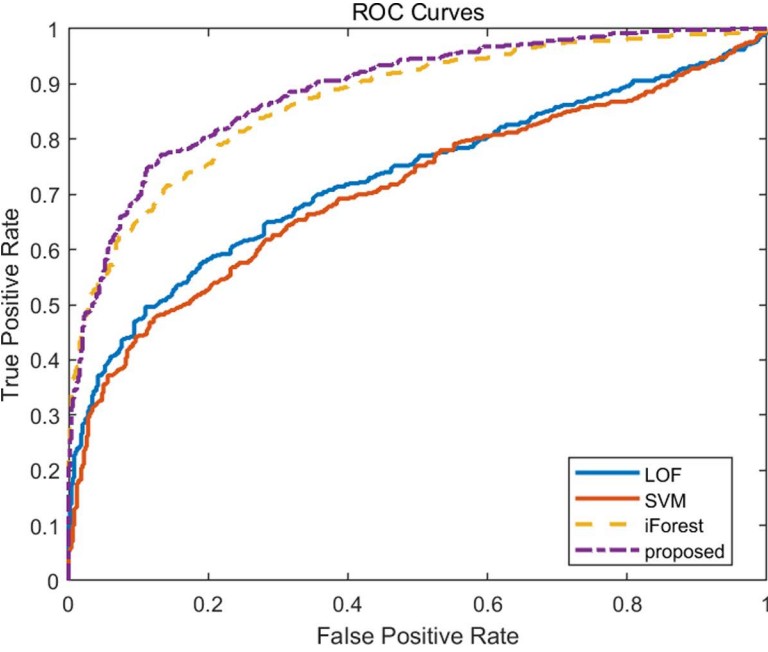

**Fig 7. Comparison of ROC curves for four algorithms with 5 nodes.**

the longer the legnth of the sliding window, the smaller the calculation error. But considering the implementation cost, a suitable value should be chosen.

The second issue is the value of crossover factor $r$. The value of the cross factor $r$ determines the number of features in the feature subset when constructing an isolated tree. When

**Table 13. The comparison of execution time of four algorithms.**

| Number of nodes | Executing time (second) | | | |
|---|---|---|---|---|
| | LOF | SVM | iForest | proposed |
| 5 | 4.6 | 6.7 | 3.5 | 3.9 |
| 10 | 18.1 | 25.8 | 5.6 | 6.1 |
| 15 | 40.9 | 45.8 | 7.1 | 8.3 |
| 20 | 77.3 | 111.4 | 9.7 | 11.2 |
| 25 | 114.5 | 180.3 | 10.4 | 12.7 |
| 30 | 185.7 | 252.9 | 11.8 | 13.9 |

the value of the cross factor $r$ is small, the number of elements in the grouped feature subset is too small, resulting in insufficient information contained in the constructed isolated tree and poor detection performance; When the value of the crossover factor $r$ is large, the isolated tree constructed by the grouped feature subset achieves balanced utilization of feature information, and the detection effect is improved; The maximum value of the crossover factor $r$ is 1.0, which means that each isolated tree is constructed using all features. The experimental results show that feature cross grouping can improve the *P, R, F1* score, and AUC of the iForest in CCL anomaly detection. The best crossover factor $r$ is 0.6 according to the experiment. The proposed algorithm can fully utilize the feature information of the data, improve the balance of feature utilization in model construction, and have better detection performance.

The third issue is the impact of data dimensionality on algorithm performance. The proposed algorithm is more suitable for high-dimensional situations. From the experiment, it can be seen that the higher the dimension, the more obvious its performance advantage. In low dimensional situations, there is no obvious advantage. The accuracy *P*, *R*, *F1* score, and AUC of the proposed algorithm are 87.79%, 88.36%, 86.30%, and 90.75% respectively when the number of nodes is 30. In terms of executing time, both the proposed algorithm and the iForest have the shortest executing time, indicating that the algorithm has low computational complexity, fast running speed, and high efficiency.

## Conclusions

In this paper, through the detailed analysis of the characteristics of multidimensional data stream, we put forward the concept of Sliding Window Model and Coherence Coefficient to describe and establish mathematical model for multi-dimensional data stream, and three main abnormal events including environment event, node event and measurement event are derived by the mathematical model, and then an improved isolation Forest algorithm is proposed for the measurement event which is the most frequent abnormal events in CCL. The algorithm combines several dimensional data with the cross factor, so that each dimension contains all the feature vectors, which overcomes the disadvantage of being easily trapped in local optimization, and does not increase a lot of computation with the sliding window. Experimental analysis shows positive results and demonstrates the system effectiveness against current three main types of current CCL vehicles. The proposed scheme in this paper can not only be used for anomaly detection in cold chain logistics, but also for anomaly detection in other multidimensional real-time data streams, especially in situations where there is strong temporal and spatial correlation between multidimensional data streams.

Although the proposed scheme and algorithm are robust to CCL, further improvements are still possible. For instance, the selection of sliding window length is obtained statistically in this experiment, and this parameter is involved in determining the accuracy of the

algorithm and the executing time. In addition, for low dimensional data streams or datasets, proposed scheme does not have significant advantages and the performance indicators do not improve much. Thus, future work will focus on further increasing the peformance of all conditions.

## Supporting information

**S1 File. coldChainDataA.** zip.
(MAT)

**S2 File. coldChainDataD** .mat.
(MAT)

**S3 File. coldChainDataF.** zip.
(MAT)

## Author contributions

**Conceptualization:** zhibo xie.

**Data curation:** Yan Luo.

**Formal analysis:** zhibo xie.

**Investigation:** zhibo xie.

**Methodology:** zhibo xie.

**Resources:** Yingjun Zhou.

**Software:** Heng Long, Chengyi Ling.

**Validation:** Yingjun Zhou, Yan Luo.

**Visualization:** Heng Long.

**Writing – original draft:** zhibo xie.

**Writing – review & editing:** zhibo xie.

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
