## [Decision Letter · Decision Letter 0]

8 Sep 2024

PONE-D-24-17883An Anomaly Detection Scheme for data stream in Cold Chain LogisticsPLOS ONE

Dear Dr. xie,

Thank you for submitting your manuscript to PLOS ONE. After careful consideration, we feel that it has merit but does not fully meet PLOS ONE’s publication criteria as it currently stands. Therefore, we invite you to submit a revised version of the manuscript that addresses the points raised during the review process.

We look forward to receiving your revised manuscript.

Kind regards,

Zhihong (Arry) Yao, Ph.D.

Academic Editor

PLOS ONE

Journal Requirements:

3. For studies involving third-party data, we encourage authors to share any data specific to their analyses that they can legally distribute. PLOS recognizes, however, that authors may be using third-party data they do not have the rights to share. When third-party data cannot be publicly shared, authors must provide all information necessary for interested researchers to apply to gain access to the data. (https://journals.plos.org/plosone/s/data-availability#loc-acceptable-data-access-restrictions) 

4) All necessary contact information others would need to apply to gain access to the data.

Reviewers' comments:

Reviewer's Responses to Questions

**Comments to the Author**

1. Is the manuscript technically sound, and do the data support the conclusions?

Reviewer #1: Yes

Reviewer #2: Yes

2. Has the statistical analysis been performed appropriately and rigorously? 

Reviewer #1: No

Reviewer #2: Yes

3. Have the authors made all data underlying the findings in their manuscript fully available?

Reviewer #1: Yes

Reviewer #2: Yes

4. Is the manuscript presented in an intelligible fashion and written in standard English?

Reviewer #1: Yes

Reviewer #2: Yes

5. Review Comments to the Author

Reviewer #1: COMMENTS

1. The abstract should provide specific numerical results from the experiments, such as precision, recall, F1 score, and AUC, to substantiate the claims of performance improvement

2. Provide more context on the challenges and limitations of current anomaly detection techniques in cold chain logistics, specifically detailing why existing methods are inadequate.

3. Expand the literature review to include recent advances and a more comprehensive analysis of existing algorithms used in cold chain logistics.

4. Provide a clearer and more detailed explanation of the improved isolated forest algorithm and the modifications made (subsampling and cross factor).

5. Ensure that all mathematical notations and equations are clearly defined and explained, with real-world examples if possible.

6. Elaborate on the derivation and significance of the correlation coefficient and how it is used to detect anomalies.

7. Describe the dataset in more detail, including its origin, size, type of data, and how it represents cold chain logistics.

8. Include a comparative analysis with more baseline algorithms and provide statistical significance testing (e.g., t-tests, p-values).

9. dd more visualizations, such as ROC curves or precision-recall curves, to illustrate the algorithm's performance across different thresholds.

10. Discuss the implications of the results in the context of real-world cold chain logistics scenarios. Include potential limitations of the proposed method and areas for future research.

11. Summarize the key contributions and explicitly state the practical impact and potential applications of the findings.

12. Review and refine the language for clarity, coherence, and conciseness. Ensure that technical terms are used consistently throughout the paper.

Reviewer #2: The paper is appropriately organized and presented.

The research method is effective in anomaly detection for data stream of Cold Chain Logistics. The experimental data is sufficiently detailed.

There are some Innovation in anomaly detection for data streams.

However, there some minor symtax problems in Lines 114,140, 154, 205, 236, 251, 471, and so on. I hope the paper could be properly revised and make it understood easily.

6. PLOS authors have the option to publish the peer review history of their article (what does this mean? ). If published, this will include your full peer review and any attached files.

**Do you want your identity to be public for this peer review?** For information about this choice, including consent withdrawal, please see our Privacy Policy .

Reviewer #1: No

Reviewer #2: No

---

## [Author Response · Author response to Decision Letter 1]

27 Oct 2024

Reviewer #1: COMMENTS

1. The abstract should provide specific numerical results from the experiments, such as precision, recall, F1 score, and AUC, to substantiate the claims of performance improvement

Thank you for your suggestion. I have added specific values for the experimental results in the abstract, including precision, recall, F1 score, and AUC.

2. Provide more context on the challenges and limitations of current anomaly detection techniques in cold chain logistics, specifically detailing why existing methods are inadequate.

I have add the challenges and limitations of current anomaly detection techniques in cold chain logistics and summarized the shortcomings of the existing technology.

3. Expand the literature review to include recent advances and a more comprehensive analysis of existing algorithms used in cold chain logistics.

I have add the literature review in the paper include recent advances and a more comprehensive analysis of existing algorithms used in cold chain logistics.

4. Provide a clearer and more detailed explanation of the improved isolated forest algorithm and the modifications made (subsampling and cross factor).

I have provided a more detailed introduction to the idea of improving the algorithm and added three core pseudocodes.

5. Ensure that all mathematical notations and equations are clearly defined and explained, with real-world examples if possible.

I have revised all mathematical notations and equations.

6. Elaborate on the derivation and significance of the correlation coefficient and how it is used to detect anomalies.

I have added explanations and derivations.

7. Describe the dataset in more detail, including its origin, size, type of data, and how it represents cold chain logistics.

I have added the dataset information.

8. Include a comparative analysis with more baseline algorithms and provide statistical significance testing (e.g., t-tests, p-values).

I think the performance indicators includes the P,R,F1 score,AUC and ROC in the paper can verify the correctness of the algorithm, and I have also studied many similar articles during this period, and their methods for verifying algorithm performance indicators are basically similar to mine.

9. add more visualizations, such as ROC curves or precision-recall curves, to illustrate the algorithm's performance across different thresholds.

I have added the ROC curves in the paper.

10. Discuss the implications of the results in the context of real-world cold chain logistics scenarios. Include potential limitations of the proposed method and areas for future research.

The experiments datasets in the paper are from real-world cold chain logistics scenarios.

I have added the potential limitations of the proposed method and areas for future research.

11. Summarize the key contributions and explicitly state the practical impact and potential applications of the findings.

I have added the key contributions and explicitly state the practical impact and potential applications of the findings.

12. Review and refine the language for clarity, coherence, and conciseness. Ensure that technical terms are used consistently throughout the paper.

I have review and refined the language and expression.

Reviewer #2: The paper is appropriately organized and presented.

The research method is effective in anomaly detection for data stream of Cold Chain Logistics. The experimental data is sufficiently detailed.

There are some Innovation in anomaly detection for data streams.

However, there some minor symtax problems in Lines 114,140, 154, 205, 236, 251, 471, and so on. I hope the paper could be properly revised and make it understood easily.

Thank you very much. I have correct the mistakes.

---

## [Decision Letter · Decision Letter 1]

25 Nov 2024

An Anomaly Detection Scheme for data stream in Cold Chain Logistics

PONE-D-24-17883R1

Dear Dr. xie,

We’re pleased to inform you that your manuscript has been judged scientifically suitable for publication and will be formally accepted for publication once it meets all outstanding technical requirements.

Kind regards,

Zhihong (Arry) Yao, Ph.D.

Academic Editor

PLOS ONE

Additional Editor Comments (optional):

Reviewers' comments:

Reviewer's Responses to Questions

**Comments to the Author**

1. If the authors have adequately addressed your comments raised in a previous round of review and you feel that this manuscript is now acceptable for publication, you may indicate that here to bypass the “Comments to the Author” section, enter your conflict of interest statement in the “Confidential to Editor” section, and submit your "Accept" recommendation.

Reviewer #1: All comments have been addressed

2. Is the manuscript technically sound, and do the data support the conclusions?

Reviewer #1: Yes

3. Has the statistical analysis been performed appropriately and rigorously? 

Reviewer #1: Yes

4. Have the authors made all data underlying the findings in their manuscript fully available?

Reviewer #1: Yes

5. Is the manuscript presented in an intelligible fashion and written in standard English?

Reviewer #1: Yes

6. Review Comments to the Author

Reviewer #1: The authors addressed all my comments. So I am satisfied with the revised manuscript. Now its in publishable form

7. PLOS authors have the option to publish the peer review history of their article (what does this mean? ). If published, this will include your full peer review and any attached files.

**Do you want your identity to be public for this peer review?** For information about this choice, including consent withdrawal, please see our Privacy Policy .

Reviewer #1: No

---

## [Editor Report · Acceptance letter]

PONE-D-24-17883R1

PLOS ONE

Dear Dr. xie,

I'm pleased to inform you that your manuscript has been deemed suitable for publication in PLOS ONE. Congratulations! Your manuscript is now being handed over to our production team.

Kind regards,

on behalf of

Dr. Zhihong (Arry) Yao

Academic Editor

PLOS ONE